# Microvesicles Released by Osteoclastic Cells Exhibited Chondrogenic, Osteogenic, and Anti-Inflammatory Activities: An Evaluation of the Feasibility of Their Use for Treatment of Osteoarthritis in a Mouse Model

**DOI:** 10.3390/cells14030193

**Published:** 2025-01-28

**Authors:** Matilda H.-C. Sheng, Charles H. Rundle, Kin-Hing William Lau

**Affiliations:** 1VA Loma Linda Healthcare System, Jerry L. Pettis Memorial VA Medical Center, Loma Linda, CA 92357, USA; matilda.sheng@va.gov (M.H.-C.S.); charles.rundle@va.gov (C.H.R.); 2Division of Biochemistry, School of Medicine, Loma Linda University School of Medicine, Loma Linda, CA 92350, USA

**Keywords:** microvesicles, osteoclasts, osteoarthritis, chondrogenesis, osteogenesis, anti-inflammation, joints, treatment

## Abstract

Extracellular vesicles (EVs), particularly exosomes (EXOs) of various skeletal and stem cells, were shown to delay osteoarthritis (OA) progression, and apoptotic bodies (ABs), another EV subtype, of osteoclasts showed osteoanabolic actions and were involved in the osteoclastic-regulation of local bone formation. Moreover, this study demonstrates that microvesicles (MVs) released by osteoclasts displayed potent pro-chondrogenic, pro-osteogenic, and anti-inflammatory activities. These activities were unique to osteoclastic MVs and were not shared by osteoclastic ABs and EXOs or MVs of other cell types. Because chronic synovial inflammation, progressive articular cartilage erosion, abnormal subchondral bone remodeling, and inability to regenerate articular cartilage are key etiologies of OA, we postulate that the foregoing activities of osteoclastic MVs could simultaneously target multiple etiologies of OA and could thereby be an effective therapy for OA. Accordingly, this study sought to assess the feasibility of an osteoclastic MV-based strategy for OA with a mouse tibial plateau injury model of OA. Briefly, tibial plateau injuries were created on the right knees of adult C57BL/6J mice, MVs were intraarticularly injected into the injured joints biweekly, and the OA progression was monitored histologically at five weeks post-injury. The MV treatment reduced the OA-induced losses of articular cartilage area and thickness, decreased irregularity in the articular cartilage surface, reduced loss of gliding/intermediate zone of articular cartilage, reduced osteophyte formation, suppressed synovial inflammation, and decreased the OARSI OA score. In summary, treatment with osteoclastic MVs delayed or reversed OA progression. Thus, this study supports the feasibility of an osteoclastic MV-based therapy for OA.

## 1. Introduction

Injuries to the joint, excessive or chronic impact loading on the joint, joint dislocations, intra-articular fractures, or genetic predisposition often cause lesions in the articular cartilage and inflammation in the synovium [1]. Chronic synovial inflammation is a key contributor to effusion, pain, chondrocyte apoptosis, and progressive breakdown of the surface layer of articular cartilage, and eventually leads to osteoarthritis (OA) [2,3]. OA is the most common debilitating degenerative joint disease, afflicting >50 million people with annual healthcare costs estimated (in 2011) to be USD 3 billion in the US alone [4], which is expected to increase by ~50% by 2040, creating a considerable socioeconomic burden [5]. A fraction of OA (~12%) is caused by acute traumatic insults to the joint and thereby is referred to as post-traumatic OA (PTOA) [6]. Primary OA affects mostly people older than 60 years of age, while PTOA can afflict younger and physically active individuals.

There is currently no cure for OA/PTOA. Mild cases are treated with medications targeting pain management and inflammation reduction [7]. Severe cases may require invasive surgical interventions [8]. All these treatment options are often disappointing, unable to reverse or halt the progression of OA, and do not yield long-term clinical benefits. As a result, the frequent eventual endpoint is joint arthroplasty or even amputation, which are not viable options for young and middle-aged patients as they may need multiple revision surgeries over their lifetime [4]. Consequently, there is an urgent need for an effective therapy that can halt or even reverse OA progression.

The pathophysiology of OA is highly complex and involves several multifactorial and interrelated etiologies. The key contributing etiologies are: (1) the chronic synovial inflammation, (2) the progressive erosion of the articular cartilage layer, (3) the abnormal remodeling/destruction of subchondral bone structure, and (4) the lack of an intrinsic ability to regenerate or repair the damaged articular cartilage. However, almost all current therapies target only a single etiology, which might be one of the major reasons they cannot yield long-term clinical benefits.

Extracellular vesicles (EVs) are nanosized, membrane-enclosed vesicles released by all cell types. They function as extracellular vehicles to transport DNA, RNA, protein, or lipid cargoes among distant cells as a form of cell-to-cell communication [9]. EVs are classified into four subtypes according to their biogenesis pathways: (1) exosomes (EXOs, 40 nm–120 nm in size) are formed by the fusion of plasma membrane and multivesicular bodies, which then release EXOs [10]; (2) microvesicles (MVs, 100 nm–1 µm in size) are formed by budding directly from the plasma membrane [11]; (3) apoptotic bodies (ABs, 50 nm–2 µm in size) are released by apoptotic cells [12]; and (4) oncosomes (1 µm–10 µm in size) are produced exclusively by malignant cells via membrane protrusion [13]. There is increasing evidence that EVs have valuable diagnostic and therapeutic applications in regenerative medicine [14,15] due to their bioactive cargo contents [16]. With respect to OA, EXOs released by synoviocytes showed great potential to be a diagnostic and monitoring biomarker for OA prognosis [16,17]. EXOs released from many skeletal cells, i.e., chondrocytes [18], synovial fibroblasts [19,20], and tendon cells [21], as well as EXOs of mesenchymal stem cells (MSCs) [22,23,24] and platelet-rich plasma [25], have also shown therapeutic potentials for OA, suggesting that EXOs released by these skeletal cells could be an excellent candidate for a novel therapy for OA.

The diagnostic and therapeutic applications of other types of EVs (i.e., MVs or ABs), especially those released by skeletal cells, have not been evaluated. In this regard, it has been shown that apoptotic osteoclasts released osteogenic ABs that appear to have key mediating roles in the local coupled bone formation [26]. Osteoblast-derived MVs contained the RANKL protein and promoted osteoclast differentiation [27]. These findings raise the interesting possibility that ABs and MVs from bone cells may also have regulatory effects on the skeleton. Our interest in osteoclastic MVs as a potential therapeutic agent for OA was triggered particularly by the intriguing findings of this study that osteoclastic MVs had potent pro-chondrogenic, anti-inflammatory, and pro-osteogenic activities. We postulate that these activities may allow the use of osteoclastic MVs to simultaneously target several key pathological hallmarks of OA, i.e., chronic synovial inflammation, loss of articular cartilage, and loss of periarticular and subchondral bones. The ability of osteoclastic MVs to target multiple key etiologies of OA could potentially yield long-term clinical benefits, which are lacking in the current therapies. Accordingly, we also performed an initial test to assess the feasibility of an osteoclastic MV-based strategy for OA. We found that biweekly intraarticular injections of MVs isolated from pooled conditioned media (CM) of osteoclastic cells into injured joints for five weeks halted or might have even reversed the progression of OA. These findings together offer strong supportive evidence for the feasibility of an osteoclastic MV-based strategy for OA.

## 2. Materials and Methods

### 2.1. Animals

Breeding pairs of C57BL/6J mice were purchased from the Jacksons Laboratory (Bar Harbor, ME, USA) and a colony of C57BL/6J mice was generated in our laboratory. Male mice were used in this study to demonstrate the feasibility of an MV-based strategy for OA. Once accomplished, results will be confirmed in female mice in the future. All animal protocols were reviewed and approved by the Institutional Animal Care and Use Committee (IACUC) of the VA Loma Linda Healthcare System and the Animal Care and Use Review Office (ACURO) of the US Army Medical Research and Materiel Command of the Department of Defense.

### 2.2. Isolation of Osteoclastic Cell-Derived EV Fractions

EV fractions were isolated by differential centrifugation of the pooled CM from osteoclast-like cells derived from RAW264.7 cells of monocytic-macrophagic lineage. RAW264.7 cells were used, because (1) they are a commonly used osteoclast progenitor cell model and can readily be induced to differentiate into osteoclast-like cells (RAW-OC) with the RANKL treatment, (2) they are more homogeneous than primary cells, and (3) they can be expanded rapidly in vitro. Briefly, the pooled CM of RAW-OC collected during the 6–7 days of RANKL treatment was first centrifuged at low speed (300× *g*) for 20 min to remove floating cells and cellular debris. The supernatant was then fractionated by sequential centrifugation steps into ABs at 800× *g* for 20 min, MVs at 16,000× *g* for 20 min, and EXOs at 100,000× *g* for 60 min fractions as described previously [28]. The identity of the MV fraction was confirmed by immunoblotting against ARF6 (an MV marker) and that of the AB fraction was confirmed by immunoblotting against TSP-1 (an AB marker). Protein content within each EV fraction was determined with the Bicinchoninic Acid (BCA) protein assay (ThermoFisher Scientific, Los Angeles, CA, USA).

### 2.3. Isolation of Murine Articular Chondrocytes

Primary chondrocytes were isolated from articular cartilage layers of both tibia and femurs of both limbs of 2- to 10-week-old male C57BL/6J mice according to ref. [29] after overnight digestion with type-2 collagenase at 37 °C and cultured in DMEM/F-12 containing 2% fetal bovine serum (FBS) and 1% insulin–transferrin–selenium (ITS) with less than 3 passages. These cells expressed *Col2α1* mRNA 554,000-fold more than *Col1α1* mRNA, and 94,600-fold more than *Col10α1* mRNA, confirming that these were bona fide articular chondrocytes [29].

### 2.4. Isolation of Murine Synoviocytes

Primary murine synoviocytes were isolated from the multiple layers of tissue adjacent to the infrapatellar fat pad (IPFP) of injured knee joints of 12- to 14-week-old male C57BL/6J mice after an intraarticular tibial plateau injury. Synovium was isolated after tibial plateau injury because the injury-induced OA reduced the size of IPFP and substantially increased the thickness of synovium from one or two layers to multiple layers of tissue. The synovium, along with part of the IPFP, was isolated from the posterior aspect of knee joints of C57BL/6J at sacrifice and minced. The minced tissues were rinsed with phosphate-buffered saline (PBS) and digested with 0.2% type-IV collagenase at 37 °C with constant shaking at 200 rpm for 60 min. The supernatant was collected after vigorous vortexing. The released synoviocytes were cultured in Synoviocyte growth medium (Cell Applications, Inc., San Diego, CA, USA) at 37 °C under an atmosphere of 5% CO_2_ with changes of medium every 3 days until confluence. Colonies of stellar cell-like synoviocytes were visible between 2 days and 2 weeks in culture. After confluent, trypsinized cells (redissolved in FBS containing 10% DMSO without culture medium) were stored in liquid nitrogen until use. The isolated synoviocytes expressed high mRNA levels of S100A4 [a marker of activated synoviocytes] and type-II collagen (Col2α1) [an extracellular protein synthesized by synoviocytes], and very low levels of type-I collagen (Col1α1) [29].

### 2.5. Isolation of Murine Osteoblasts

Primary mouse osteoblasts were isolated from calvaria of 1-week-old C57BL/6J mice and were cultured as described previously [30]. Proliferation was assayed by the 5-Bromo-2-deoxyuridine (BrdU) incorporation assay in MC3T3-E1 cells or primary murine osteoblasts as described in [31,32]. Osteoblastic differentiation was monitored by determining the cellular alkaline phosphatase (ALP) activity normalized as the total cellular protein as previously described [31]. Mineralized nodule formation assay using primary mouse marrow cells was performed as described in [31].

### 2.6. In Vivo Bone Formation Assay

A calvarium injection assay [33] was used to evaluate the in vivo bone formation activity of the osteoclastic MV fraction. Osteoclastic MV (~60 µg MV protein in 20 µL PBS) or PBS was injected at the surface of the calvarium of male C57BL/6J mice daily for 5 consecutive days. Bone formation was assessed by the calcein/demeclocycline double labeling approach as described previously [30]. Calcein and demeclocycline were administered through IP on day 1 and day 5, respectively. All mice were then sacrificed 2 days later and dynamic histomorphometric bone formation parameters were measured on the external surface of the injected calvaria.

### 2.7. A Mouse Model of Closed Intraarticular Tibial Plateau Injury Model of OA

A validated closed intraarticular tibial plateau compression loading-induced injury model of OA [34] was used. The injury was produced on the intraarticular tibial plateau using an Instron servohydraulic mechanical tester under isoflurane inhalation anesthesia. The right knee was positioned on a support attached to the load cell of a mechanical tester. The excursion of a blunt indenter blade was set on the top of the tibial plateau, and an impact force was applied under a defined force of 55 N at a speed of 200 N/s. This impact force effectively caused significant injuries to the synovium, meniscus, and articular cartilage but did not create tibial plateau fractures. The mice were sacrificed at 5 weeks post-injury.

### 2.8. Histology

Both the injured right and contralateral uninjured left joints (containing distal femoral and proximal tibial ends) were fixed with 10% cold neutral buffered formalin for 3–4 days, rinsed with PBS, and stored in PBS. The joints were decalcified, dehydrated, and embedded in paraffin by standard procedures. Serial 5-µm sagittal thin sections were cut from the medial side. Multiple serial sections corresponding to the injured tibial plateau region were deparaffinized, followed by graded ethanol dehydration, and stained with safranin O/fast green for cartilage or 0.4% toluidine blue in sodium acetate buffer. The total area and the thickness of the articular cartilage layer around the impacted tibial plateau, and the average depth of the gliding and immediate zones at the afflicted joint on at least three sections of each joint were measured with the OsteoMeasure™ (SciMeasure Analytical Systems, Decatur, GA, USA). The articular surface smoothness was quantitated by a semi-quantitative scoring system according to ref. [35].

### 2.9. Assessment of Relative Severity of OA

The severity of OA in PBS-injected untreated and MV-treated injured knee joints was determined in a blinded fashion with the OARSI OA Cartilage Histopathology Assessment method [36] based on a scale from 0 to 6, with 6 representing the most severe and 0 representing no OA.

### 2.10. Statistical Analyses

Normal distribution and homogeneity of variance of the data sets were confirmed with Levene’s test. Statistical significance between the two test groups was analyzed with a two-tailed Student’s *t*-test. Results are shown as mean ± SEM. The difference was considered significant when *p* < 0.05.

## 3. Results

### 3.1. MVs but Not ABs or EXOs Released by Osteoclasts Released Exhibited Chondrogenic Activity

Treatment of articular chondrocytes with osteoclastic MVs for 24 h significantly upregulated the expression of chondroblastic marker genes, i.e., *Acan*, *Prg4*, *Col2α1*, *Col10α1*, *Sox9* (Figure 1A) and, in the presence of 2% fetal calf serum (FCS), promoted the formation of acidic proteoglycan-producing colonies (Figure 1B) compared to the PBS-treated control cells (PBS was the solvent for the resuspension of all EV fractions), indicating that osteoclastic MVs have pro-chondrogenic activity. The chondrogenic activity of osteoclastic MVs was also compared to that of osteoclastic AB and EXO fractions isolated from the same pooled CM. Treatment with all EV fractions at the same EV protein concentration for 24 h each upregulated the expression of *Prg4* on chondrocytes, but the stimulatory effects of MVs were 1- and 2-fold greater than that of ABs and EXOs, respectively (Figure 1C). Osteoclastic MVs also increased the formation of acidic proteoglycan-producing colonies after culturing for 21 days in a chondrogenic medium in the presence of 10% FCS (Figure 1D). In contrast, osteoclastic AB was slightly but not significantly increased, whereas osteoclastic EXOs not only did not enhance but significantly reduced the ability of chondrocytes to form acidic proteoglycan-producing colonies, confirming that osteoclastic MVs, but not ABs or EXOs, have potent pro-chondrogenic activity in chondrocytes.

### 3.2. Osteoclastic MVs Exhibited Anti-Inflammation Activity

To evaluate if osteoclastic MVs possess anti-inflammation activity on synoviocytes, we determined the effects of the 48 h MV treatment on the expression level of genes associated with synovial inflammation, i.e., *Igf1*, *Cd44*, *pPBP*, and *Mmp9* in mouse synoviocytes. Figure 2A shows that osteoclastic MV treatment of primary synoviocytes reduced cellular expression levels of each test gene by ~65% to ~80% compared to PBS-treated controls, suggesting that osteoclastic MVs exhibited potent anti-inflammatory activity in synoviocytes. Figure 2B shows that osteoclastic MV, AB, and EXO fractions all showed strong suppressive effects, as each EV fraction significantly reduced the expression level of the test synovial inflammatory genes compared to PBS-treated control cells, but the suppressive effect was the largest for EXOs and the smallest for ABs.

### 3.3. Osteoclastic MVs, but Not ABs or EXOs, Had Osteogenic Activity

We next determined the effects of osteoclastic MV treatment on BrdU incorporation (as a measure of proliferation) into osteoblasts isolated from the calvarial bones of C57BL/6J mice after 24 h of incubation, cellular ALP activity (as an index of osteoblastic differentiation) after 48 h of incubation and mineralized nodule-forming activity of mouse osteoblasts after 21 days of incubation. MVs isolated from pooled CM of mouse osteoclasts significantly increased BrdU incorporation (Figure 3A) and cellular ALP activity (Figure 3B) of MC3T3-E1 osteoprogenitor cells. In contrast, MVs of MC3T3-E1 cells did not increase BrdU incorporation (Figure 3A) or ALP activity (Figure 3B), while MVs isolated from pooled CM of human HEK-293 kidney cells increased BrdU incorporation (Figure 3A) but had no effect on ALP activity (Figure 3B) in MC3T3-E1 cells. Because osteoclastic MVs, but not MVs of osteoblastic and kidney cells, promoted both osteoblastic proliferation and differentiation, the osteogenic activity of MVs is probably unique to MVs of osteoclastic cells. Osteoclastic MVs also enhanced the mineralized nodule-forming activity of primary osteoblasts (Figure 3C), confirming the bone formation stimulatory activity of osteoclastic MVs. We next compared the osteogenic activity of osteoclastic MVs with that of osteoclastic ABs or EXOs. All three EV fractions increased BrdU incorporation (Figure 3D), but only MVs significantly increased cellular ALP activity (Figure 3E). Because increased osteoblast proliferation without an increase in osteoblastic differentiation would not lead to an increase in bone formation [32], it follows that osteoclastic MV, but not EXO or AB fractions, have osteoanabolic activity.

A calvarium injection assay [33] was then used to evaluate the in vivo bone-forming activity of osteoclastic MV. Accordingly, daily local injection of osteoclastic MVs at ~60 µg MV protein (in 20 µL PBS) for five consecutive days significantly increased dynamic formation parameters, i.e., TLS/B.Pm, MAR, BFR/B.Pm, at the external layer compared to PBS-injected control calvaria (Figure 4), confirming that osteoclastic MVs are a potent local osteoanabolic agent in vivo.

### 3.4. Biweekly Intraarticular Injections of Osteoclastic MVs Slowed Down or Reversed OA Progression in a Mouse Intraarticular Tibial Plateau Injury Model of OA/PTOA

We next tested whether osteoclastic MVs could be an attractive candidate for OA therapy using our recently validated mouse closed intraarticular tibial plateau injury model of OA/PTOA [34] with a biweekly intraarticular injection regimen that was used previously for EphA4-based signaling therapeutic strategy [35]. Accordingly, a dose (~60 µg MV protein in 20 µL PBS) of MVs isolated from pooled CM of RAW-OC was injected intraarticularly into the injured joint one day after the injury and every two weeks thereafter, and the severity of OA on injured joints at five weeks post-injury was assessed as previously described [34,35].

Toluidine blue staining of acidic cartilage on longitudinal sections of uninjured contralateral joints and injured joints treated with PBS or MV shows that the PBS-injected injured joint had less and thinner articular cartilage layer (yellow arrows), more osteophyte-like tissues (red arrows), and greater mineralized area in the meniscus (black arrows) than the uninjured contralateral joint. However, the MV treatment appeared to return the articular cartilage thickness similar to that of a contralateral uninjured joint. It also showed much less osteophyte-like tissues and reduced area of mineralized meniscus (Figure 5A). Quantitative measurements of the average surface and average articular cartilage thickness at the injured tibial plateau revealed that PBS-injected injured knee joints lost significant articular cartilage surface area (Figure 5B) and reduced average thickness (Figure 5C) compared to the contralateral uninjured tibial plateau, confirming the development of OA at five weeks after the tibial plateau injury. In contrast, the MV-treated injured joints had greater articular cartilage area and thicker cartilage layer than PBS-injected OA joints, which were not statistically different from those of uninjured joints, suggesting that the treatment might have halted or even reversed the progression of OA development.

Losses of surface smoothness and the gliding/intermediate zones of articular cartilage are well-established characteristics of OA. Therefore, we determined the relative surface smoothness or irregularity and the average thickness of the gliding/intermediate zones of the articular cartilage layer at the site of the injury in the tibial plateau using a semi-quantitative approach [34]. The tibial plateau injury treated with MVs had increased smoothness (or less irregular and shorter surface length) of the articular cartilage surface (Figure 5D, left side) and displayed smaller loss of the gliding/intermediate zone (Figure 5D, right side) than PBS-injected injured joints. Similarly, the MV-treated injured joints had a significantly lower average area of osteophyte-like tissues at and around the injured tibial joint than the untreated (PBS-injected) injured joints (Figure 5E). The relative severity of OA in the PBS-injected untreated and the MV-treated injured joints was then assessed with the semi-quantitative histology-based OARSI OA Cartilage Histopathology Assessment System [36,38]. The biweekly MV treatment substantially reduced the OARSI OA score in the injured joint (Figure 5F). Thus, osteoclastic MV treatment lessened the severity of OA that was developed in response to intraarticular tibial plateau injuries.

Synovial hyperplasia is a key pathologic hallmark of OA. Figure 6 shows that the tibial plateau injury greatly increased the thickness of the synovium layer at PBS-injected (untreated) injured joints (indicated by red arrows) compared to that in the uninjured contralateral tibia joint. Conversely, the osteoclastic MV treatment appeared to reverse synovial hyperplasia in the injured joint and returned the thickness of the synovium layer to that of contralateral uninjured joints. Synovial hyperplasia is induced largely through the actions of pro-inflammatory cytokines as a result of chronic synovial inflammation. Thus, these findings are consistent with the concept that osteoclastic MVs have potent anti-inflammatory activities in the synovium of the OA joint in vivo.

## 4. Discussion

This study demonstrates for the first time that MVs isolated from pooled CM of osteoclastic cells displayed potent chondrogenic activity in chondrocytes, osteogenic activity in osteoblasts, and anti-inflammatory activity in synoviocytes. Among the three major forms of EVs released by osteoclastic cells, only MVs had this unique ability to promote all three biological activities. Unlike osteoclastic MVs, MVs isolated from the pooled CM of HEK-293 kidney cells or MC3T3-E1 osteoblastic cells failed to promote the proliferation and/or differentiation of osteoblastic cells. Thus, we tentatively conclude that the coexistence of the three foregoing activities is unique to osteoclastic MVs and not ABs or EXOs. These exciting findings may have clinical implications. Accordingly, chronic synovial inflammation, inability to repair/regenerate the damaged articular cartilage, and the loss of subchondral bone are three key etiologies of OA/PTOA. Thus, the biological activities of osteoclastic MVs could render them an excellent candidate for an effective OA therapy that could simultaneously address multiple etiologies of OA. It is reasoned that if such an osteoclastic MV-based therapy could indeed simultaneously target three major pathologies of OA, this type of therapy could potentially yield long-term and meaningful clinical benefits.

This study also evaluated the effects of biweekly intraarticular injections of osteoclastic MVs into injured joints on the progression of OA development using our previously validated mouse closed intraarticular tibial plateau compression loading-induced injury model of OA [34] as an initial test of the feasibility of an osteoclastic MV-based strategy for OA. The tibial plateau injury model of OA was chosen because it is non-invasive and does not involve surgical opening of the joint. This would not only allow preservation of the integrity of the synovial environment and avoidance of manipulation of the articular tissues but also avoid any issues concerning potential infection due to the surgical procedure. The tibial plateau was chosen for the site of injury, because (1) this site has thicker articular cartilage than do other joints, (2) the tibial plateau is one of the most common sites of traumatic insults at the lower extremity of PTOA (or OA) patient population with a bimodal age distribution [39] and is thereby clinically relevant, and (3) the intraarticular tibial plateau injuries in the mouse resemble the most common traumatic insults to the knee joints encountered in PTOA patients [39]. The compression loading-induced injury approach is the most convenient and direct noninvasive means to induce substantial injuries to the tibial plateau, and the intraarticular injury to the synovium, the meniscus, and the articular cartilage surface induced by the compression loading has led to consistent and progressive development of OA/PTOA (seen in over 99% of the injured joints), as it consistently caused OA/PTOA in the injured joint at five weeks post-injury [34]. Moreover, this mouse model was used successfully by us to demonstrate that the EphA4-signaling-based strategy could be an effective therapy for OA/PTOA [35].

This study clearly shows that biweekly osteoclastic MV intraarticular injections one day after tibial plateau injury for five weeks substantially reduced the OA-associated loss of average articular cartilage area and decreased the OA-induced decrease in average articular cartilage thickness, reduced the OA-associated increase in the irregularity of the articular cartilage surface and the loss of gliding/intermediate zones of articular cartilage, and drastically reduced the large increase in the OA-induced formation of osteophyte and reversed the OA-associated thickening of synovial membrane (an index of synovial inflammation). As a result, there was a large reduction in the OARSI OA disease score. Together, these findings strongly suggest that the osteoclastic MV treatment might have slowed down or even reversed the progression of OA, and thereby they offer strong support for the feasibility of an osteoclastic MV-based therapy for OA.

We should emphasize that only a single dose of osteoclastic MVs (60 µg MV protein, which was selected randomly) was tested in this study because this study was our first and preliminary attempt to obtain evidence for the feasibility of osteoclastic MV-based therapeutic strategy for OA and related skeletal diseases. In the future, we will perform a dosage study to determine an optimal dosage and to more fully characterize the dose-dependent anti-OA effects of osteoclastic MVs.

The nature and identity of the active component(s) of osteoclastic MVs responsible for the chondrogenic, osteogenic, and anti-inflammatory activities are not known. It is also not clear whether these activities are due to a protein, lipid, RNA, or DNA molecule or if these activities were attributed to a single or multiple components. Information about the identity of the active components of the MVs is critically important for the determination of an effective dosage range and the proper development and optimization of the osteoclastic MV-based therapy. We hypothesized that the active component(s) of osteoclastic MVs for the foregoing activities are one or more transmembrane proteins on the MV. This assumption was based on the fact that MVs are formed by budding directly from the plasma membrane [11], which is enriched in plasma membrane-associated proteins, many of which are functional membrane-associated growth factors, growth factor receptor-ligands, as well as receptors for various cytokines, all of which are signal transduction mediators/initiators. It is our hypothesis that membrane-associated growth factors or receptor ligands on osteoclastic MVs would bind to their respective cognate cell surface receptors or ligands of nearby osteoblastic and chondrogenic precursors to exert the respective osteogenic and chondrogenic actions, respectively. Likewise, the membrane-associated cytokine receptors on MVs could act as decoy receptors for inflammatory cytokines to prevent their inflammation actions on nearby synoviocytes or related cells. This intriguing hypothesis is worthy of further investigation.

It has, however, been reported that EXOs secreted by certain skeletal cells and MSCs has exhibited disease-modulating actions against OA progression in various animal models [18,19,20,21,22,23,24,25,40,41]. Investigations into the potential mediators of their anti-OA activities have suggested that many of their OA modulation activities could be mediated by non-coding RNAs, such as lncRNAs and miRNAs. For example, the chondroprotective effects of the MSC-derived EXOs have been proposed to be mediated by the *KLF3-AS1 lncRNA* [42]. EXOs released by chondrogenic progenitor cells have also been reported to alleviate OA symptoms through *miR221-3p* [43]. EXOs released by IPFP MSCs halted OA progression by alleviating articular cartilage degradation through the actions of *miR100-5p* [44]. EXOs released by synovial MSCs delayed OA progression through the *miR155-5p*-mediated increases in chondrocyte proliferation and migration and reduction in chondrocyte apoptosis [45], and EXOs of umbilical cord MSCs acted in part through *LncRNA H19* to improve OA symptoms and relieve pain in a rat cartilage defect-induced OA model [46]. While MVs also carried miRNAs, the miRNA content in MVs was relatively depleted compared with those in its cells of origin [47]. MVs also contained certain novel noncoding RNAs that had no known functions [48]. Consequently, we will not overlook the possibility that at least some of the OA-modifying activities of osteoclastic MVs could also be mediated through non-coding RNAs. Our future studies will examine this interesting possibility.

This study has several limitations that need further investigation. First, the osteoclastic MV fraction isolated by differential centrifugation was not pure with unknown amounts of contaminating impurities, including ABs and other cellular components. We chose the differential centrifugation approach to isolate MVs from pooled CM because this method was convenient, easy to perform, and offered the highest yield of MVs compared to other purification methods. Nevertheless, some of these contaminants could be biologically active and thereby may confound the results and interpretations. Accordingly, some of the key findings need to be confirmed with more purified preparations of osteoclastic MVs.

Second, this study utilized an administration regimen of biweekly intraarticular injections of osteoclastic MVs. This regimen, while effective, might not be the most efficacious strategy. To determine an optimal administration regimen, it is necessary to know approximately how long the injected osteoclastic MVs would stay inside the injured joint after intraarticular administration. This information allows not only the determination of whether a single administration would be sufficient but also of the frequency of as well as the time between intraarticular administrations. The biological effects were measured up to after five weeks of treatment. It would be essential to assess the long-term efficacy to determine if the therapy was still efficacious after long-term treatment. It would also be important to determine whether the therapeutic effects would be sustained after cessation of the therapy.

Lastly, while this osteoclastic MV-based strategy showed promising protective effects on OA or delayed the progression of OA when it was administered one day post-injury, it is not clear if this strategy would still be effective in slowing down or reversing the disease progression after OA has already been established. This information is essential since OA patients usually seek medical intervention only after OA has been well-established and severe symptoms developed.

## 5. Conclusions

This study demonstrates that the isolated MV fraction from pooled CM of osteoclastic cells displayed potent pro-chondrogenic activity on chondrocytes, pro-osteogenic activity on osteoblasts, and anti-inflammatory activity on synoviocytes, although it remains unclear whether these activities were derived from a single or multiple components of the MV fraction or whether they were derived from protein, lipid, RNA, or DNA molecules. Nevertheless, this study showed that osteoclastic MVs when administered at an early stage of the joint injury have protective actions for OA development and progression, and as such it offers strong supporting evidence for the feasibility of an osteoclastic MV-based therapeutic strategy to prevent or treat OA.

## Figures and Tables

**Figure 1 cells-14-00193-f001:**
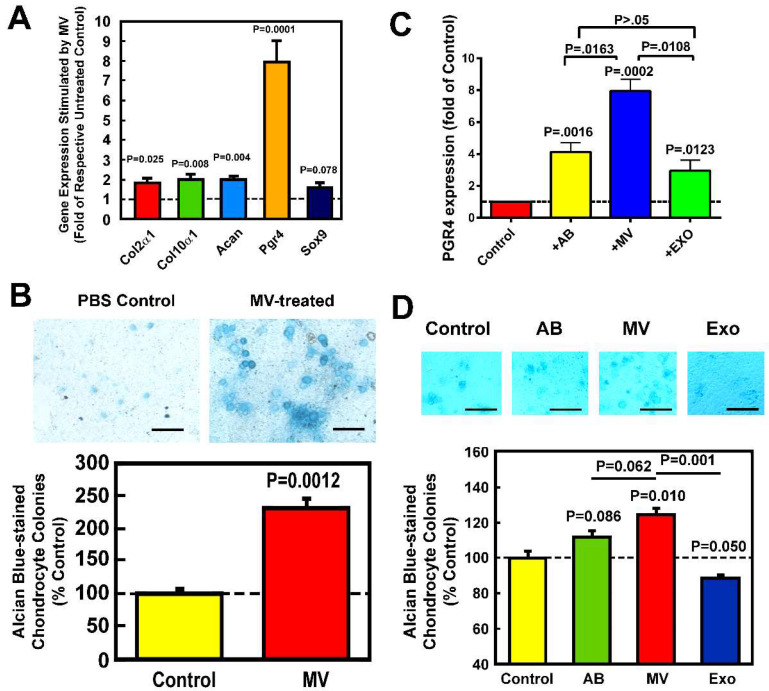
Osteoclastic MVs, ABs, and EXOs upregulated the expression of chondrogenic genes (**A**), especially *Pgr4* (**C**), but only MVs and not ABs or EXOs significantly promoted the formation of acidic proteoglycan-producing colonies (**B**,**D**) in primary chondrocytes. Osteoclastic MVs, ABs, and EXOs were isolated from the same pooled CM of RAW-OC by sequential centrifugation and were redissolved in PBS. A: Articular chondrocytes were treated with 45 µg MV protein/mL of osteoclastic MVs (or PBS as control) for 24 h, and the expression levels of chondrogenic genes (i.e., *Col2α1*, *Col10α1*, *Acan*, *Pgr4*, and *Sox9*) were determined by RT-qPCR and normalized against *β-actin* (mean ± SEM, n = 3). Dashed line = 100% of controls. B: Chondrocytes cultured in DMEM/F-12 medium containing 2% FCS and 1% ITS were treated with PBS or osteoclastic MVs for 21 days. Top: acidic proteoglycan-producing colonies were stained with alcian blue. Scale bar = 1 mm. Bottom: the alcian blue stain was eluted with 8M guanidine HCl and absorbance read at 600 nm (mean ± SEM, n = 3). Dashed line = % of the PBS control. C: Chondrocytes were treated with osteoclastic MVs, ABs, or EXOs at a concentration of ~45 µg protein/mL, or PBS for 24 h. The expression level of *Pgr4* mRNA in cells treated with each EV was determined by RT-qPCR normalized against *β-actin* mRNA. Results are reported as relative % of respective PBS-treated cells (mean ± SEM, n = 4). The *p*-values are determined by two-tailed Student’s *t*-tests. D: Chondrocytes were treated with MVs, ABs, EXOs, or PBS in DMEM/F-12 medium with 10% FCS for 21 days. Top: acidic proteoglycan-producing colonies were stained with alcian blue. Scale bar = 1 mm. Bottom: the eluted alcian blue stain. Results were shown as mean ± SEM (n = 3). Dashed line = % of the PBS control. The level of stimulation by MVs in panel D was smaller than that shown in panel B because 10% FCS instead of 2% FCS was used. The *p*-values are determined by two-tailed Student’s *t*-tests.

**Figure 2 cells-14-00193-f002:**
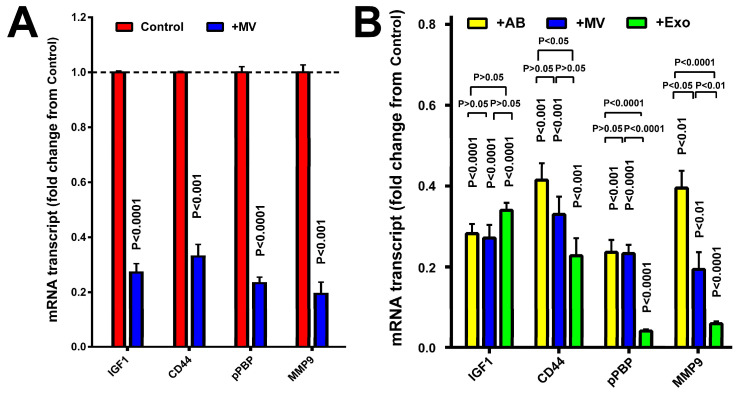
Osteoclastic MVs (**A**), as well as osteoclastic ABs and Exos (**B**) substantially downregulated the expression of pro-inflammatory genes in primary mouse synoviocytes. Primary mouse synoviocytes were treated with osteoclastic MVs or PBS in (**A**) or osteoclastic MVs, ABs, EXOs, or PBS (**B**) for 48 h. Total RNA was extracted from all treated cells, and the mRNA levels of several pro-inflammatory genes, i.e., *IGF-1*, *CD44*, *pPBP*, and *MMP9* were determined by RT-qPCR and normalized against *β-actin* mRNA (mean ± SEM, n = 3). The *p*-values were determined by two-tailed Student’s *t*-tests compared to the respective control group. The dashed line is 100% of the respective controls.

**Figure 3 cells-14-00193-f003:**
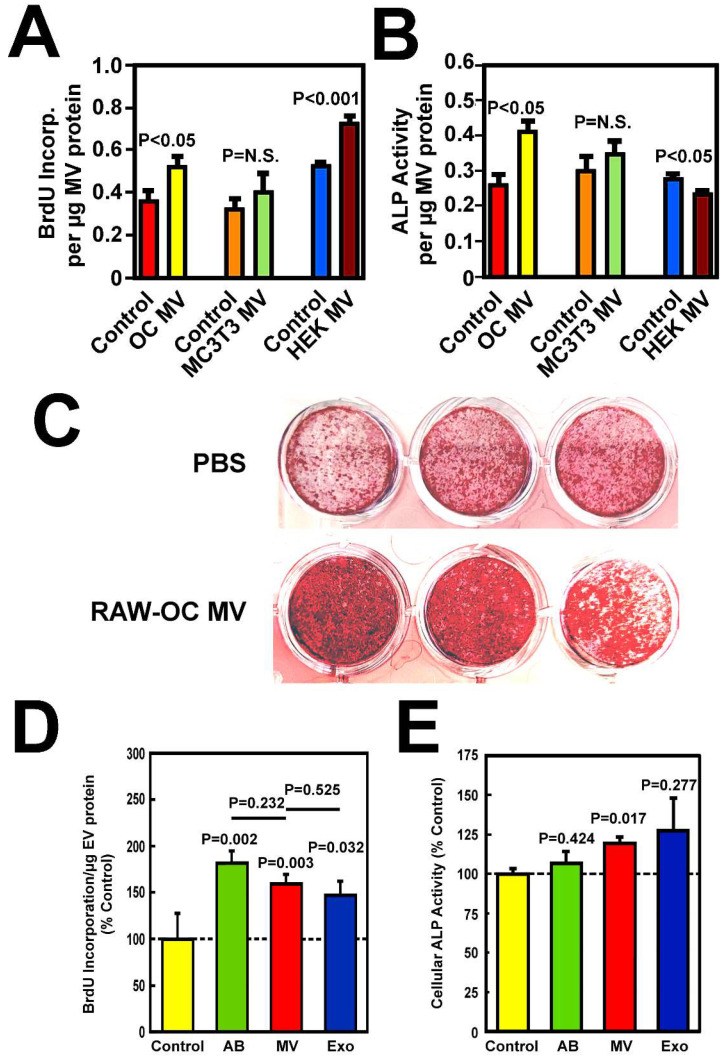
MVs isolated from pooled CM of osteoclasts (OC), but not MC3T3-E1 or HEK cells, increased both BrdU incorporation (**A**) and cellular ALP activity (**B**) in MC3T3-E1 cells. MVs of RAW-OC also promoted mineralized nodule formation after 21 days of incubation (**C**) in primary osteoblasts. (**A**,**B**): MVs were added to MC3T3-E1 cells for 24 h (left) or 48 h (right). BrdU incorporation and ALP activity were each normalized against MV protein content. Mean ± SEM, n = 6. (**C**): Comparison of the relative intensity of alizarin red staining after extraction of dye with acetic acid and measured spectrophotometry at 405 nm as described in [37] indicates that MVs of RAW-OC increased mineralized nodule formation by ~75%. (**D**,**E**): Osteoclastic MVs, but not ABs or EXOs, stimulated both BrdU incorporation and cellular ALP activity in primary mouse osteoblasts. (**D**): The BrdU incorporation (reported as % control, mean ± SEM, n = 6). (**E**): The cellular ALP specific activity (mean ± SEM, n = 3). The *p*-values are determined by two-tailed Student’s *t*-tests. Dashed lines in both panels = 100% of corresponding controls.

**Figure 4 cells-14-00193-f004:**
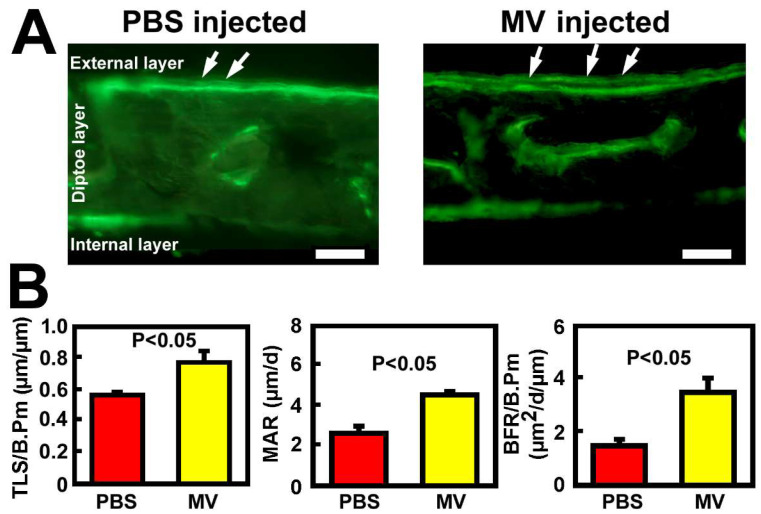
Intracranial injections of osteoclastic MVs for 5 days increased local de novo bone formation at 14 days. (**A**). Fluorescent photomicrograph of a vertical cross-sectional view of a representative PBS-injected and a representative osteoclastic MV-injected mouse calvarium. Calcein/demeclocycline double-labeled surfaces at the external layer are indicated by arrows. Bar = 50 µm. (**B**). Dynamic histomorphometric bone formation parameters. Results are shown as mean ± SEM, n = 3 per group. The *p*-values were determined by two-tailed Student’s *t*-tests.

**Figure 5 cells-14-00193-f005:**
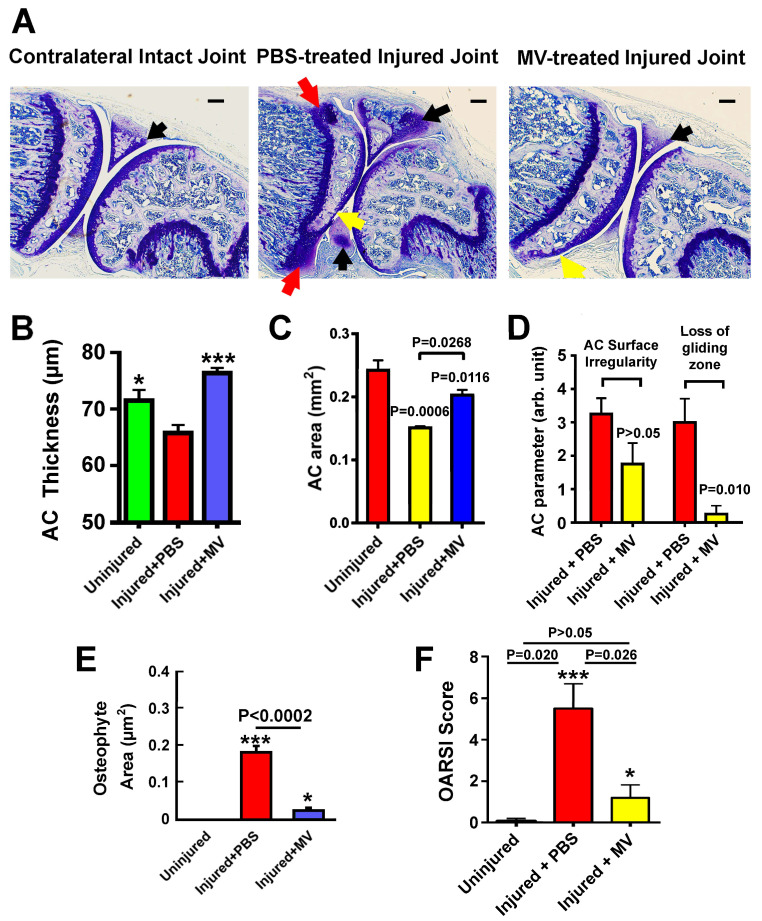
Biweekly intraarticular injections of osteoclastic MVs have beneficial effects on the prevention of OA. (**A**): Toluidine blue staining of acidic cartilage on thin sagittal sections of a representative contralateral uninjured tibial joint and an injured joint of each group treated with PBS or MVs. Yellow arrows = areas with thinning of articular cartilage layer; black arrows = mineralized meniscus; and red arrows = osteophyte-like tissue. Scale bar = 100 µm. (**B**): Average articular cartilage area on the injured tibial plateau site, measured on multiple sections of each mouse. (**C**): Average thickness of articular cartilage on injured tibial joints, measured on multiple sections of each animal. Measurements of average articular cartilage area and thickness were performed with the OsteoMeasure system. Results are shown as mean ± SEM (n = 4). (**D**): Smoothness of articular cartilage and gliding/intermediate zone area measurements were performed as previously described [34,35]. Results are shown as mean ± SEM (n = 4). (**E**): Average area of osteophyte-like tissues on tibial joints was measured on multiple sections of each animal. Results are shown as mean ± SEM (n = 4 mice). (**F**): The relative severity of OA of PBS-treated and osteoclastic MV-treated injured joints, along with contralateral uninjured joints, were each assessed with the 6-point scale OA cartilage histopathology-based OARSI OA scoring system. Statistical analyses were performed by two-tailed Student’s *t*-tests, and *p*-values were compared to intact uninjured joints. * *p* < 0.05 and *** *p* < 0.001.

**Figure 6 cells-14-00193-f006:**
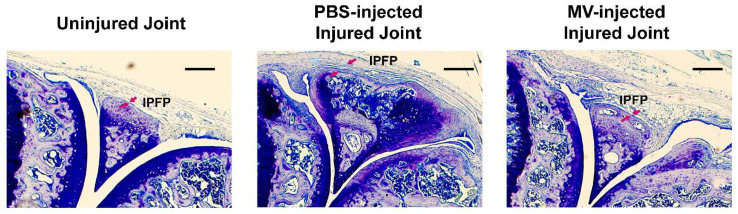
Biweekly intraarticular injections of osteoclastic MVs reduced synovial hyperplasia after intraarticular tibial plateau injury. Thin sagittal sections of joint sections of knee joints were stained with toluidine blue. The red arrows indicate the thickness of the synovium. This figure shows that the intraarticular tibial plateau injury after five weeks substantially increased the thickness of the synovium at the injured joint, and the osteoclastic MV biweekly injections prevented or reduced the synovial hyperplasia. IPFP = infrapatellar fat pad. Scale bar = 100 µm.

## Data Availability

The research data that support the findings of this study are stored at an approved storage facility within the Loma Linda VA Healthcare system and are available from other investigators for review upon reasonable request and approval by the Loma Linda VA Healthcare system.

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
