# Peer review of "Microvesicles Released by Osteoclastic Cells Exhibited Chondrogenic, Osteogenic, and Anti-Inflammatory Activities: An Evaluation of the Feasibility of Their Use for Treatment of Osteoarthritis in a Mouse Model"

_cells, 2025, doi:10.3390/cells14030193_

Round 1
Reviewer 1 Report
Comments and Suggestions for Authors
Major revisions are required and please include missing supplementary material. Proper characterization of Microvesicles and differentiation from other EV subtypes (AB and Exo) is needed to make claims regarding distinct effect of MVs versus other EV subtypes. Please explain the rationale behind choosing a dosage of 60ug and whether other doses were tested before settling. Furthermore, introduction is lacking the rationale for testing microvesicles and differentiating current literature of MVs versus EV subtypes in terms of therapeutic capability.
Author Response
Major revisions are required and please include missing supplementary material.
Our response: We have significantly revised our manuscript to address all concerns raised by the Editor and the Reviewers. However, we are puzzled by the criticism of missing supplementary materials since our manuscript does not include supplemental data.
Proper characterization of Microvesicles and differentiation from other EV subtypes (AB and Exo) is needed to make claims regarding distinct effect of MVs versus other EV subtypes.
Our response: The identity of MV fractions was confirmed by immunoblotting against ARF6 (a MV marker), and that of AB fraction was confirmed by immunoblotting against TSP-1 (an AB marker). This information is included in the Methods section (lines 114-116).
Please explain the rationale behind choosing a dosage of 60 ug and whether other doses were tested before settling.
Our response: This is our first and preliminary attempt to provide preliminary evidence for the feasibility of osteoclastic MV-based therapeutic strategy for osteoarthritis and related skeletal diseases and did not intend to fully characterize the anti-OA effects of osteoclastic MV. Accordingly, we only chose a single dose of Mv for testing in this study. We chose the initial dosage of 60 µg randomly. Because it showed promising results, this dosage was used for the rest of this study. In our future studies, a dosage experiment will be performed to establish an optimal dosage.
Furthermore, introduction is lacking the rationale for testing microvesicles and differentiating current literature of MVs versus EV subtypes in terms of therapeutic capability.
Our response: In response to this criticism, we added a stronger rationale for testing MV as therapeutic agent for OA in the Introduction (lines 76-94). The discussion on the comparison of the use of other EV fractures, particularly Exo of other skeletal cells and stem cells, for treating OA, with osteoclastic MV is included in the fourth paragraph of the Discussion section (lines 421-438).
Reviewer 2 Report
Comments and Suggestions for Authors
The manuscript describes the characterization of micro-vesicles (MV) obtained from RAW264 or primary bone marrow cell osteoclasts for potential chondrogenic, osteogenic, and anti-inflammatory activities. Only basic outcomes assessments were made, but the available data indicates that the osteoclast derived MV do possess the capacity to influence chondrocyte, osteoblast, and synoviocyte gene expression.
In a second experiment, the investigators used the osteoclast MV to treat a novel mouse model of knee OA in mice. The model entails creating an injury to the tibia plateau and while the investigators indicate that model reliable produces OA, that is unclear. Consequently, understanding how effective use of the osteoclast MV to prevent OA in these injured knees becomes somewhat problematic. The investigators treated the knees 1 day after inducing this injury and then again at 2 and 4 weeks before killing mice for analysis of knee OA at 5 weeks. The reported data indicate that the MV did substantially reduce OA progression and would be in line with the anti-inflammatory effects also noted when tested on synoviocytes.
On the whole, the experimental results support the conclusions drawn by the investigators. The premise of the paper includes a novel approach to use of MV and provides interesting insight to the activities of osteoclasts.
Points to be addressed:
Line 14: sentence is not clear. Which cell types show increased cell proliferation?
95: Please state ages of mice for the different experiments. Please comment on why only male mice were used.
127: change to “multiple layers of tissue”
135: change to “cell-like synoviocytes”
138: the mRNAs for type I collagen are Cola1 and Col1a2 and for type II collagen is Col2a1, please correct and identify which were tested.
145: normalized to what? “as cellular protein” or “to total cellular protein”. Please clarify.
191: P<0.05.3. ?
195: Genes should be italicized.
226 and 233: Please clarify what media and serum amount was used when performing these experiments.
Figure 2A: Unclear why panel A is needed when data are also presented in panel B.
253: chondrogenic should be inflammatory
Figure 3: use of the MC3T3-E1 and HEK293 cells is OK, but a better control would have been RAW264 cells without RANKL treatment.
282: please explain why ALP and BrdU incorporation were normalized against MV protein content rather than total cellular protein (or DNA).
284: how was 75% measured?
307: 66 ug ? Everywhere else it was 60, please confirm 66 is correct.
Author Response
The manuscript describes the characterization of micro-vesicles (MV) obtained from RAW264 or primary bone marrow cell osteoclasts for potential chondrogenic, osteogenic, and anti-inflammatory activities. Only basic outcomes assessments were made, but the available data indicates that the osteoclast derived MV do possess the capacity to influence chondrocyte, osteoblast, and synoviocyte gene expression.
Our response: No response is needed.
In a second experiment, the investigators used the osteoclast MV to treat a novel mouse model of knee OA in mice. The model entails creating an injury to the tibia plateau and while the investigators indicate that model reliable produces OA, that is unclear. Consequently, understanding how effective use of the osteoclast MV to prevent OA in these injured knees becomes somewhat problematic. The investigators treated the knees 1 day after inducing this injury and then again at 2 and 4 weeks before killing mice for analysis of knee OA at 5 weeks. The reported data indicate that the MV did substantially reduce OA progression and would be in line with the anti-inflammatory effects also noted when tested on synoviocytes.
Our response: The development of this noninvasive intraarticular tibial plateau compression loading-induced injury mouse model of OA/PTOA has been published in Calcif Tissue Int (2020) 106:158-171 (https://doi.org/10.1007/s00223-019-00614-0). We used this model because this model is noninvasive and does not involve surgically opening of the joint. This would not only allow preservation of the integrity of the synovial environment and avoidance of manipulation of the articular tissues, but also would avoid any issues concerning potential infection due to the surgical procedure. We chose the tibial plateau site for the site of injury because this site has thicker articular cartilage than do other joints and because tibial plateau is one of the most common site of traumatic insults at lower extremity of PTOA (or aged OA) patients. The compression loading-induced injury approach is the most convenient and direct means to induced substantial injuries to the tibial plateau that eventually lead to OA/PTOA. We found that the intraarticular injury to the synovium, the meniscus, and the articular cartilage surface induced by the compression loading have led to consistent progressive development of OA/PTOA (seen in over 99% of the injured joints). Importantly, we have successfully used this mouse model to demonstrate that the EphA4-signaling-based strategy could be developed for prevention and treatment of OA/PTOA. This is published in J Bone Miner Res (2022) 37:660-674 (https://doi.org/10.1002/jbmr.4500). Consequently, we believe that this mouse model of knee OA is appropriate and suitable for this study.
On the whole, the experimental results support the conclusions drawn by the investigators. The premise of the paper includes a novel approach to use of MV and provides interesting insight to the activities of osteoclasts.
Our response: No response is necessary.
Line 14: sentence is not clear. Which cell types show increased cell proliferation?
Our response: We apologize for the confusion. What we meant was an increased osteoblast proliferation. However, as required by the Academic Editor, the Abstract is now completely rewritten.
95: Please state ages of mice for the different experiments. Please comment on why only male mice were used.
Our response: A colony of C57BL/6J mice were generated in our laboratory and were used in this study. The age of mice in each experiment is now clearly indicated. The reason why only male mice were used in this study was because it was our first and preliminary study to demonstrate feasibility of an osteoclastic MV-based therapeutic strategy for OA/PTOA. We intend to perform more in-depth studies to characterize the therapeutic effects and compare the therapeutic effects in male versus female mice to determine if there is a sex-dependent difference in the responses to osteoclastic MV in the future. This information is added to the revised manuscript (lines 96-99).
127: change to “multiple layers of tissue”
Our response: It has been changed as recommended (line 131).
135: change to “cell-like synoviocytes”
Our response: The correction has been made as suggested (line 138).
138: the mRNAs for type I collagen are Cola1 and Col1a2 and for type II collagen is Col2a1, please correct and identify which were tested.
Our response: The mRNA for type 1 collagen was Colα1 and type II collagen was Col2α1. This information is added to the revised manuscript (lines 141-142).
145: normalized to what? “as cellular protein” or “to total cellular protein”. Please clarify.
Our response: We revised the manuscript to indicate that the ALP activity was normalized to total cellular protein (i.e., specific activity) (line 148).
191: P<0.05.3. ?
Our response: It should be P<0.05 (line 187). We thank the Reviewer pointing out our typo.
195: Genes should be italicized.
Our response: The name of the genes is now italicized (lines 191-192, also line 212-213 and line 220).
226 and 233: Please clarify what media and serum amount was used when performing these experiments.
Our response: The media and amount of serum in each experiment are added to the legend of original Fig. 1 (now Fig. 2) (line 214 and 222).
Figure 2A: Unclear why panel A is needed when data are also presented in panel B.
Our response: We apologize for the apparent confusion. Panels A and B of original Fig. 2 (now Fig. 3) were two different sets of experiment addressing two different questions. Panel A was aimed to demonstrate that osteoclastic MV was effective in suppressing expression of inflammatory genes in primary mouse synoviocytes. After that was accomplished, we then performed the second experiment to compare the relative effects of osteoclastic MV, AB, and Exo (at the same EV protein level) on the suppression of inflammatory gene expression in primary mouse synoviocytes. The results of this second experiment are shown in panel B. We show both panels because it is clear from panel A that MV was effective. Conversely, panel B gives a better and direct comparison of the anti-inflammatory effects of MV with those of AB and Exo.
253: chondrogenic should be inflammatory
Our response: It should have been anti-inflammation and not chondrogenic. We have corrected this error and thank the Reviewer for pointing out our oversight (lines 234).
Figure 3: use of the MC3T3-E1 and HEK293 cells is OK, but a better control would have been RAW264 cells without RANKL treatment.
Our response: Our objective for original Fig. 3A and B (now Fig. 4A and B) was to determine whether the enhanced osteogenic effects were specific for osteoclastic MV and not for MV released by other osteoclastic cell types. MC3T3-E1 and HEK293 cells were chosen in our study as non-osteoclastic cell type. Accordingly, while we appreciate and understand the Reviewer’s comment, we feel that undifferentiated RAW264 cells would not be an appropriate control for this particular experiment.
282: please explain why ALP and BrdU incorporation were normalized against MV protein content rather than total cellular protein (or DNA).
Our response: We assumed that the active component(s) in MV was integral transmembrane protein(s) based on the rationale provided in the third paragraph of the Discussion (lines 402-420). Accordingly, we normalized the biological effects (e.g., ALP and BrdU incorporation) against MV protein content for normalization purpose.
284: how was 75% measured?
Our response: We apologize for the omission. After Alizarin red staining, the color of the mineralized nodules was extracted with acetic acid and then measured at 405 nm spectrophometrically in a microplate reader. The ~75% increase was based on an increase in an ~75% absorbance in MV-treated cells compared to control cells. This information is added to the revised manuscript (lines 272-274).
307: 66 ug ? Everywhere else it was 60, please confirm 66 is correct.
Our response: It should be 60 µg (line 297). We again thank the Reviewer for finding our typo.